# A Novel Three-Polysaccharide Blend In Situ Gelling Powder for Wound Healing Applications

**DOI:** 10.3390/pharmaceutics13101680

**Published:** 2021-10-14

**Authors:** Chiara Amante, Tiziana Esposito, Pasquale Del Gaudio, Veronica Di Sarno, Amalia Porta, Alessandra Tosco, Paola Russo, Luigi Nicolais, Rita P. Aquino

**Affiliations:** 1Department of Pharmacy, University of Salerno, Via Giovanni Paolo II 132, I-84084 Fisciano, SA, Italy; camante@unisa.it (C.A.); tesposito@unisa.it (T.E.); vdisarno@unisa.it (V.D.S.); aporta@unisa.it (A.P.); tosco@unisa.it (A.T.); paorusso@unisa.it (P.R.); aquinorp@unisa.it (R.P.A.); 2Materias s.r.l., University of Naples, “Federico II” Campus San Giovanni a Teduccio, I-80146 Naples, Italy; nicolais@materias.it

**Keywords:** alginate, pectin, chitosan, doxycycline, wound dressing, in situ forming hydrogel, exudate absorbance, MMP-9 inhibition

## Abstract

In this paper, alginate/pectin and alginate/pectin/chitosan blend particles, in the form of an in situ forming hydrogel, intended for wound repair applications, have been successfully developed. Particles have been used to encapsulate doxycycline in order to control the delivery of the drug, enhance its antimicrobial properties, and the ability to inhibit host matrix metalloproteinases. The presence of chitosan in the particles strongly influenced their size, morphology, and fluid uptake properties, as well as drug encapsulation efficiency and release, due to both chemical interactions between the polymers in the blend and interactions with the drug demonstrated by FTIR studies. In vitro antimicrobial studies highlighted an increase in antibacterial activity related to the chitosan amount in the powders. Moreover, in situ gelling powders are able to induce a higher release of IL-8 from the human keratinocytes that could stimulate the wound healing process in difficult-healing. Interestingly, doxycycline-loaded particles are able to increase drug activity against MMPs, with good activity against MMP-9 even at 0.5 μg/mL over 72 h. Such results suggest that such powders rich in chitosan could be a promising dressing for exudating wounds.

## 1. Introduction

Lesions and ulcers caused by trauma, thermal injuries, and surgery affect a large part of the population with a strong economic impact on the world healthcare system [1,2]. The incidence of chronic wounds is continuously growing due to the incessant population aging and its consequent health issues as cardiovascular diseases or diabetes [3,4]. The healing of such wounds involves interaction among different cells, signaling molecules, and extracellular matrix components and the process can be easily impaired and altered by pathogens and microorganisms, leading to severe wound and systemic infections, especially in aged people [5,6,7]. To treat chronic or bacteria-infected wounds, in the last years, numerous dressings have been developed based on natural polymers as the main components of several devices with particular reference to alginate, collagen, chitosan, pectin, and hyaluronic acid due to their biocompatibility, biodegradability, and for the ability to stimulate the healing process [8,9,10,11,12].

An ideal wound dressing should be biocompatible and biodegradable while being able to adapt to different shapes and depths of the wound, as well as to be removed without causing pain. Moreover, it should ensure good transpiration of the wound through the diffusion of oxygen and water vapor and remove the excess fluid for exudating wounds [13,14,15]. Exudate, maintaining a proper wound environment, stimulates the healing process, but when in excess, it can slow down the healing process, also becoming an excellent substrate for bacteria proliferation [16,17].

Polysaccharide-based hydrogels, due to their ability to manage exudates, can be used as dressings to prevent wound dehydration [18,19,20]. However, most dressings based on natural polysaccharides lack conformability and exudate absorbance [21,22]. In situ forming carbohydrate-based hydrogels can solve several issues related to wound managing due to their ability to properly fill the wound site, absorbing and retaining the excess of exudate [23,24], while the use of a suitable mixture of polysaccharides allows the formation of a transparent dressing, enabling the evolution of healing process [25,26].

In a previous work [27], we demonstrated the possibility of using alginate-pectin blends to produce in situ gelling microparticulate powders with proper wound dressing properties in terms of adhesiveness and transpiration that are able to quickly gel when in contact with wound fluids. 

Alginate, an anionic hydrophilic linear polymer composed of β-D-mannuronic acid and a-L-guluronic acid, is one of the most-used natural polymers for wound dressing due to its biocompatibility, biodegradation, and absorption properties, as well as its ability to control the release of a drug [28,29,30]. Moreover, alginate, with high mannuronic content, is able to induce cytokine production by human monocytes [31], while amidated pectin with a low degree of methylation is able to speed up the gelling rate [32]. Moreover, blends of alginate with other natural polysaccharides, such as pectin, showed better properties in wound healing applications [27].

Chitosan, a linear cationic polysaccharide, is composed of β-(1-4)-linked D-glucosamine and N-acetyl-D-glucosamine, formed by N-deacetylation of chitin [33], is used in drug delivery and wound dressing [34,35,36,37,38]. Chitosan is able to induce macrophages activation and accelerate wound healing [39], and it has an antimicrobial effect that depends on its molecular weight, concentration, and microbial species under examination [40], and it has a hemostatic property due to the interaction between the positive charge of chitosan and negative charge on cell membranes of the red blood cells [41]. Moreover, due to the protonation of its amino groups that allow it the reduction of the pH of the inflammatory area, chitosan is considered a polymer with analgesic activity [42].

The aim of this work has been the development of in situ gelling microparticle powders loaded with doxycycline, which are converted into gel after administration onto the wound bed. Alginate/pectin and alginate/pectin/chitosan blends have been studied. The polymer blend ratio, with particular attention to the amount of chitosan, has been evaluated considering its influence on the properties of the particles and on drug release when used as a wound care formulation.

Doxycycline has been used as an antimicrobial model drug due to its wide antibacterial spectrum against Gram-positive and Gram-negative bacteria [43] and its ability to inhibit host matrix metalloproteinases (MMPs), hyper expressed in chronic wounds through the chelation of calcium or zinc ions necessary for the enzymatic activity of metalloproteinases [44,45].

The in situ gelling powders produced by mini spray drying technology have been characterized in terms of yield, morphology, size distribution, FT-IR analysis, and drug encapsulation efficiency. Furthermore, the fluid uptake ability of the powders and drug release profiles have been studied. Finally, antimicrobial tests against *Staphylococcus aureus*, MTT test and ELISA assay for cytotoxic and pro-inflammatory activity, and zymography for the inhibition of MMPs, using human keratinocyte (HaCaT cells), were performed in order to assess the safety and efficacy of the particles when used to treat infected wounds.

## 2. Materials and Methods

Doxycycline monohydrate was obtained from Carbosynth (Compton, UK). Sodium alginate Kelton LVCR from brown algae (1% viscosity 35 mPa s; mannuronic/guluronic ratio 70/30) was kindly donated by Dompè S.p.A (Dompè Farmaceutici S.p.A. L’Aquila, Italy). Pectin Amid CF 025 D (amidated low methoxyl grade, degree of esterification 23–28%, degree of amidation 22–25%, molecular weight 120 kDa) has been kindly donated by Herbstreith&Fox (Herbstreith & Fox GmbH & Co. KG, Neuenbürg, Germany). Chitosan with a low molecular weight (1% viscosity in acetic acid 20–80 mPa s; 75–85% deacetylated) was purchased from Sigma-Aldrich (Milan, Italy). Acetic acid used for the dissolution medium was obtained from Sigma-Aldrich, Italy. Acetonitrile for HPLC (VWR Chemicals, Briare, France), Water for HPLC (VWR Chemicals, Briare, France), and orthophosphoric acid 99% (Carlo Erba Reactifs, Cornaredo (MI), Italy) were used as the mobile phase for high-performance liquid chromatography (HPLC).

### 2.1. Powders Preparation

#### 2.1.1. Alginate/Pectin and Doxycycline Powders

The alginate/pectin solution was prepared by dissolving both polymers in distilled water. The total concentration of polymers was set between 0.15% and 0.50% (*w*/*v*) and alginate/pectin ratios (1:1, 1:3, 3:1) were considered.

All formulations were processed by Mini Spray Dryer B-290 (BuchiLaboratoriums-Tecnik, Flawil, Switzerland) with optimized parameters: aspirator 100%, drying airflow 560–580 L/h, air pressure 6 atmospheres, feed rate 5 mL/min, 120 °C inlet temperature, 65–68 °C outlet temperature, and a nozzle diameter of 0.5 mm. The powders obtained were characterized in terms of production yields, calculated as the ratio between the amount of product obtained and the total amount of the material processed.

#### 2.1.2. Alginate/Pectin/Chitosan, and Doxycycline Powders

Powders were produced by mini spray drying starting from feed solutions that differ in composition and polymer ratio. Different ratios of alginate/pectin/chitosan (1:1:1, 1:1:3 and 1:1:7), total concentration of polymers (0.15% *w*/*v*), and doxycycline concentration (2% and 1% *w*/*w* of polymers) were set. Feeds were obtained by adding doxycycline to an aqueous solution of alginate/pectin/chitosan under gentle stirring. The alginate/pectin solution was prepared by dissolving alginate and, after complete dissolution of it, the pectin in distilled water under vigorous stirring for 1 h while chitosan was dissolved in an acidic aqueous solution (1% *v/v* CH_3_COOH) under gentle stirring overnight. The alginate/pectin solution was added, slowly and with the use of Ultra-Turrax^®^ T25 (IKAWorks GmbH & Co., Staufen, Germany), to the acid solution of chitosan.

### 2.2. Drug Content and Encapsulation Efficiency

Drug content was calculated as the ratio between actual doxycycline weight and powder weight, while encapsulation efficiency (E.E.) was calculated as the ratio of actual to theoretical drug content [46]. About 5 mg of the different formulations was stirred into 5 mL of PBS buffer (0.1 M, pH 7.4) to obtain a uniform dispersion. The dispersion was shortly sonicated, centrifuged (6000 rpm for 10 min) to remove polymers, and the supernatant obtained was filtered (filters of 0.45 µm) and analyzed by HPLC (Slavica M. Sunarìc et al., 2013). The Agilent 1100 Series instrument (Agilent 1100 Series HPLC, Agilent Technologies, Santa Clara, CA, USA) with a UV detector, set at 349 nm, was used for the HPLC analysis. The separation was performed on a reversed-phase column (Kinetex XB-C18, 100A, 75 × 2.10 mm, 2.6 µm, Phenomenex, Torrance, CA, USA), using a mixture of acetonitrile (A) and water with a pH of 2.5 with orthophosphoric acid (B) as the mobile phase. The volume ratio of solvent A against solvent B was 22:78, the flow rate was 0.5 mL/min, and the injection volume was 20 µL. The results were compared with a calibration curve in a range of concentrations between 0.5 and 34.0 µg/mL.

### 2.3. Powders Characterization 

#### 2.3.1. Morphology and Particle Size Distribution

The morphology of all powders was observed by scanning electron microscopy (SEM), using a Carl Zeiss EVO MA 10 microscope with a secondary electron detector (Carl Zeiss SMT Ltd., Cambridge, UK) and with 20 KeV accelerating voltage. To verify the uniformity of the particles, no less than 20 SEM images were taken. The powders were dispersed on an aluminum stub (Agar Scientific, Stansted, UK) and were coated with a thick gold layer (200–400 Å) (LEICA EMSCD005 metallizator) before microscopy.

Particle size distribution and mean diameter were evaluated by static light scattering (SLS) (N5, Beckman Coulter, Miami, FL, USA), taking the average of three measurements for each sample. Each formulation was diluted in dichloromethane (DCM) and sonicated three times for 10 min, with and without the presence of a surfactant (1% *v*/*v* between in DCM).

Results were expressed as mean diameter d10, d50, d90, indicating the volume diameters at 10th, 50th, and 90th percentiles, respectively.

#### 2.3.2. Fluid Uptake Ability

Franz diffusion cell was used to determine the fluid uptake ability of powders. About 8 mg of powder was placed on a previously weighed membrane disc filter (PES, 0.45 µm, 25 mm, Pall Corporation, Port Washington, NY, USA). The membrane was lying down on the vertical receptor compartment of the Franz cell, filled with simulated wound fluid (SWF), composed of 50% fetal calf serum (Sigma Aldrich, Milan, Italy) and 50% maximum recovery diluent (Sigma Aldrich, Milan, Italy) consisting in 0.1% (*w*/*v*) peptone, a peptic digest of animal tissue, and 0.9% (*w*/*v*) sodium chloride [47,48]. The SWF was thermostated at 37 °C, and during the experiment, the receptor compartment was filled with SWF to maintain a constant fluid volume in the compartment. At regular time intervals, the weight of the gel was measured after removing the membrane from the receptor until a constant weight was observed [49].

The swelling ratio of each formulation was determined as the ratio between the weight of the swollen gel (W_s_) and the weight of the dried powder producing the gel (W_d_), using the following Equations [50]:Swelling = W_s_/W_d_(1)

All experiments were performed at least in triplicate.

#### 2.3.3. Fourier Transform Infrared Spectroscopy (FT-IR) 

FT-IR analysis was evaluated using an FT-IR spectrophotometer (IRAffinity-1S, Shimadzu Corporation, Kyoto, Japan) equipped with a MIRacle ATR accessory with a ZnSe crystal plate. The samples were put on the crystal plate and analyzed in the spectral range of 4000 to 600 cm^−1^, using 128 scans with a 1 cm^−1^ resolution step.

### 2.4. In Vitro Drug Release

The release of doxycycline encapsulated in microparticles was evaluated using Franz-type vertical diffusion cells (15 mm, 7 mL, Hanson research corporation, Chatsworth, CA, USA), which allows the keeping of the powders in a humid environment similar to those encountered at the wound site. About 20 mg of the formulation was placed on the donor side, fully covering the membrane that was inserted between the donor and the receptor compartment (PES, 0.45 µm, 25 mm, Pall Corporation, Port Washington, NY, USA). The receptor phase was filled with SWF and thermostated at 37 °C, and magnetically stirred at 250 rpm. At the set time intervals, aliquots (250 µL) were withdrawn from the receptor compartment and replaced with an equal volume of the fresh buffer solution. The drug released was identified by HPLC as previously described (see Section 2.3.3), and all release experiments were performed in triplicate.

### 2.5. Rheological Studies 

Rheological data of completely swollen hydrogels were acquired with an Anton Paar MCR-102 Rheometer fitted with plate-plate geometry (PP25 with a diameter of 24,985 mm). The hydrogels were placed between the flat plate and the platform heated at 37 °C. The distance between the plates was determined according to the stickiness of the gels tested. Amplitude sweep tests were conducted, setting the strain amplitude in the range 0.01–200% and angular frequency at ω = 10 rad/s. Experiments were performed in triplicate.

### 2.6. Antimicrobial Assay

To evaluate the antibacterial efficacy of formulations, both time-kill studies and disk diffusion assay were conducted.

Time-kill studies were performed as previously described [51]. Briefly, an overnight culture of S. aureus strain A170 was diluted 30-fold with prewarmed Mueller-Hinton Broth (MHB) and incubated at 35 °C under constant shaking. The late-log phase of growth was normalized to 2.5 × 10^8^ cells/mL, and 200 μL were used to inoculate three sterile 96-well plates further treated with the different APCs formulations corresponding to 0.50 mg/mL of doxycycline used. At defined time points (1, 2, 3, and 7 days), viable counts were calculated to give the CFU/mL (colony-forming unit per ml) by plating serial dilutions on MHA incubated at 35 °C. Kill curves were plotted with time against the number of CFUs recovered. Each antimicrobial assay was performed in triplicate on separate days.

“Disc diffusion assay” was carried out on Mueller–Hinton agar (MHA) according to the Clinical and Laboratory Standard Institute (CLSI) guidelines, with suitable modifications [52]. Briefly, sterile cotton swabs were dipped into a bacterial suspension (10^8^ cells/mL) and then spread on an MHA agar plate. To test the antimicrobial activity of the different formulations, S. aureus (ATCC 6538), which normally colonizes wounds, has been studied. The plate was spotted with powders diluted with lactose to test different concentrations of doxycycline encapsulated using pure doxycycline as the control. After incubation at 37 °C for 24 h, the zone of inhibition was determined by measuring the diameter of the clearing zone around the powders.

### 2.7. In Vitro Test

#### 2.7.1. Cell Culture Conditions for MTT and ELISA Assay

The human keratinocyte cell line HaCaT (Cell Line Services, Eppelheim, Germany) was cultivated using Dulbecco’s modified Eagle’s medium (DMEM) supplemented with 10% fetal bovine serum, with antibiotics (10,000 U/mL penicillin and 10 mg/mL streptomycin), and with growth factors contained in Keratinocyte-SFM following the instructions reported in [53]. The cells were maintained at 95% humidity and 37 °C in an atmosphere of 5% CO_2_ and were serially passed at 70/80% confluence.

#### 2.7.2. MTT Assay

To assess the potential cytotoxicity of particles, HaCaT cells were treated with particles made of alginate, pectin, chitosan, and alginate/pectin/chitosan blends (0.01–10 µg/mL) in triplicate for 24 h. Cell viability was determined with the colorimetric MTT (3-[4,5-dimethylthiazol-2-yl]-2,5-diphenyltetrazolium bromide) assay. A total of 25 mL of MTT was added, and the cells were incubated for another 3 h. Subsequently, the cells were, lysed and the formazan was solubilized with 100 mL of an aqueous solution containing 50% N,N-dimethylformamide and 20% SDS at a pH of 4.5. The amount of formazan was quantified spectrophotometrically at 550 nm with a microplate reader (Titertek Multiskan MCC/340, LabSystem).

For statistical analysis of the data, the One Way Analysis of Variance (ANOVA) followed by Bonferroni’s multiple comparison post-test was used.

#### 2.7.3. ELISA Assay

To evaluate the pro-inflammatory activity of powders, pro-inflammatory cytokines, and chemokines, IL-6, TNFα and IL-8 were measured in cell-free supernatants using commercial enzyme-linked immunosorbent assay kits (ELISAs). HaCaT cells were treated as reported MTT assay, using the same polymers and concentrations to correlate the results obtained for cell viability with the release of pro-inflammatory cytokines.

Statistical differences of the data were assessed with One Way Analysis of Variance (ANOVA) followed by Bonferroni’s multiple comparison post-test or Tukey’s post-test.

#### 2.7.4. Cell Culture and Treatments for Zymography

Human Epidermal Keratinocytes adult cell line (HEKa) were grown in an EpiLife^®^ Medium supplemented with Human Keratinocyte Growth Supplement (HKGS) (Gibco), at 37 °C, in a 5% CO_2_ atmosphere, without serum. In fact, as serum contains gelatinases, for gelatin zymography, it is important to obtain serum-free conditioned media [54]. For experiments, trans-well filters (12 mm in diameter with 0.4 µm pore polycarbonate membrane insert, Corning) were used. Cells were seeded on trans-well filters at a density of 1 × 10^5^ cells/well and allowed to reach confluence over 24 h. Before treatment with the hydrogel, the cells were washed in PBS, and EpiLife^®^ Medium without supplement was added to the upper compartment of all wells. The lower chambers of different wells were filled with hydrogels with different doxycycline amounts: 0.5, 1, and 5 μg in a total volume of 1 mL. One well, with EpiLife^®^ Medium without supplement, was used as a negative control. The conditioned media in the upper compartments were taken after 24, 48, and 72 h of treatment, centrifuged, and stored at −80 °C until subsequent uses.

#### 2.7.5. SDS-PAGE Gelatin Zymography

SDS-PAGE gelatin zymography was carried out to detect and quantitate gelatinases in the conditioned media from HEKa cultures after treatments according to a previously reported method [54]. Briefly, 10% polyacrylamide gel containing 0.1% gelatin was loaded with 20 μL of each supernatant mixed with SDS-PAGE sample buffer without boiling or reducing. After electrophoresis, to remove SDS, the gels were washed in 0.25% (*v*/*v*) Triton X-100 for 30 min at room temperature with gentle stirring and then incubated at 37 °C overnight in a digestion buffer (50 mM Tris-HCl [pH 7.4], 150 mM NaCl, 10 mM CaCl_2_, and 0.01% Brij-35) to allow gelatin digestion. Gels were stained with Coomassie Brilliant Blue G-250 (0.5% (*v*/*v*), Coomassie blue in 30% (*v*/*v*) methanol, and 10% (*v*/*v*) acetic acid) and de-stained with 10% (*v*/*v*) acetic acid and 25% (*v*/*v*) methanol until clear bands indicating proteolytic activity became visible. Image analysis was performed by the Image Quant LAS 4000 (GE Healthcare, Waukesha, WI, USA) digital imaging system and quantified by Image Quant TL software.

## 3. Results and Discussion

### 3.1. Preparation and Characterization of the Powders

In order to evaluate the influence of chitosan on the characteristics of particles, alginate/pectin and alginate/pectin/chitosan in situ gelling powders with different ratios were produced by spray drying. Preliminary studies were conducted to optimize the process parameters and the operating conditions for feed solution preparation when chitosan was included in the polysaccharides blend. In fact, due to the formation of a polyelectrolyte complex between the protonated amines of the chitosan and the carboxylate groups of the alginate [55,56,57], the feed was limited in its final pH and concentration. The optimal concentration of the polymers in terms of feasibility and yield of the process was found at 0.15% (*w*/*v*), whereas higher concentrations led to the formation of a coalescence feed resulting in low process yield (<25%). As reported in Table 1, a comparison between mini spray-dried particles obtained from alginate/pectin (AP) and alginate/pectin/chitosan (APC) at different ratios shows that the process yield was not significantly affected by the presence of chitosan when the lowest amount of the polymer was used to prepare the feeds. However, when feeds were prepared with the higher amount of chitosan, a slight increase in process yield was registered, moving from about 60% to 70%, probably due to an increase in viscosity of the feed related to alginate-chitosan interaction able to promote crosslinking of the polymer chains during the formation of the particles [58,59]. The loading of doxycycline did not significantly affect either the size or the morphological properties of the particles, as shown in Figure 1d–f.

Drug encapsulation efficiency (E.E.) was also influenced by the presence of chitosan in the polymeric blend. In fact, alginate/pectin particles showed an E.E. of about 70% not significantly affected by the drug/polymers ratio, whereas the E.E. for alginate/pectin/chitosan particles ranged between 67% and 78%, depending on the relative amount of chitosan in the polymer blend. Such results could suggest that an interaction between amino residues of the chitosan and hydroxyl groups of doxycycline might take place as the chitosan concentration increases in the feed.

Reported in Figure 1 are the SEM images of powders obtained with different polysaccharides blends. AP particles show a mean diameter ranging between 2.07 and 2.85 µm, depending on the polymer ratio. Smaller particles were obtained when a 1:1 polymer ratio was used, whereas the largest particles were produced processing feeds with the highest amount of pectin. Size distribution and particle roundness were also affected by the presence of pectin in the blend. In fact, AP-13 particles exhibited wide size distribution, a tendency to agglomerate, and a partially shrunk shape that might be related to the presence of xylose, galactose, and arabinose in the structural side chains of the pectin that leads to a reduction of the interactions between the polymers and consequently to a collapsed particle structure after the drying process [60].

In APC powders, the amount of chitosan influenced both particle size and size distribution. Higher polymer concentrations led to smaller particles (see Table 1), while doxycycline did not affect the particle size and morphology of the particles. Indeed, AP particles were characterized by a smooth surface, whereas the addition of chitosan led to particles exhibiting a rough surface; such a modification could be attributed to the precipitation of chitosan on the matrix of alginate and pectin during the drying phase due to its lower solubility in the feed. In fact, surface roughness increased accordingly to the amount of chitosan used to prepare the feed, as it could be recognized by a lower tendency to form agglomerates (see Figure 2a–c).

In order to evaluate the ability to absorb exudate by the in situ-forming gel when spread on a wound, the fluid uptake ability was studied using a simulated wound fluid (SWF) to mimic the wound bed environment in vitro. All formulations demonstrated quick interaction with SWF, with APCs formulations were considerably faster than powders without chitosan. In fact, APC-117 powders were able to move into a gel within 2 min while adapting to the surface on which they were spread (see Figure 3).

Figure 4 shows the fluid uptake of different powders, demonstrating that all APC formulations were able to reach the maximum uptake in about 10 min, whereas APs needed about 15 min to reach the same status. However, it is possible to observe that fluid absorption kinetics are strongly dependent on the concentration of chitosan. In fact, although the formulation APC-117 had a slower uptake, it was able to absorb higher amounts of SWF compared to other APC powders, around 50% and 60% more than APC-113 and APC-111, respectively. This phenomenon can be explained by taking into account that APC-117 particles exhibit a smaller size and higher surface area due to the very wrinkled surface and a less ordered structure consisting of chitosan macromolecules on the surface of the polymers blend matrix [61]. Doxycycline-loaded APCs formulations exhibited lower fluid uptake ability compared to their blank homologs. Such a reduction was correlated to the amount of doxycycline loaded into the particles. In fact, APCx-1D formulations exhibited a higher fluid uptake than APCx-2D. These results are inconsistent with the hypothesis that doxycycline interacted with chitosan via hydrogel bonding or the formation of new bonds, reducing the ability of the particles to interact with water molecules.

FT-IR studies on doxycycline and polymer blend particles were conducted to verify the formation of any interaction between the polymers during the formation of the particles, as well as between the drug and the polymer matrix after the production of the loaded particles. In Figure 5A, the spectra of the particles obtained by a single polymer and particles obtained by alginate/pectin/chitosan blends with different polymer ratios are reported. As reported in previous work [49], blank alginate particles (Figure 5A-a) presented two characteristic bands at 1602 and 1408 cm^−1^ related to the antisymmetric and symmetric stretching vibration of the alginate carboxyl group, respectively. Blank pectin particles also presented the characteristic amide I and amide II bands at 1675 and 1590 cm^−1^, while COO stretching was observed at 1418 cm^−1^ (Figure 5A-b). Chitosan particles (Figure 5A-c) exhibited the amide I band at 1655 cm^−1^ and the bands related to the amide II and NH bending partially overlapped at 1552 cm^−1^, as well as a band associated with CH_2_ bending at 1375 cm^−1^ and the band related to antisymmetric C-O-C and C-N stretching at 1149 cm^−1^. APC particles presented several bands of the single polymers that overlapped. However, COO bands of alginate, amide stretching of pectin, and the NH bending band of chitosan shifted at lower wavenumbers according to the increase of chitosan in the blend (Figure 5A-d–f), namely ranging from 1528 to 1521 cm^−1^ for the characteristic chitosan band and between 1402 and 1396 cm^−1^ for the alginate COO bands.

Panel B reports the spectra of the drug (Figure 5B-g), blank polymer particles, and loaded particles comprise the region ranging between 1800 and 1300 cm^−1^, where doxycycline shows the most characteristic peaks. In fact, as also reported in Silva et al. [62], at 1680 cm^−1^, the band representing the C=O of the amide group is present, while at 1602 and 1573 cm^−1^, the carbonyl groups of the two rings are present. At 1525 cm^−1^, the band corresponding to the amino group of the amide is shown, whereas the band at 1455 cm^−1^ represents the C=C skeleton vibration. 

The spectrum of alginate/pectin/chitosan particles loaded with doxycycline (see Figure 5B-i) shows the changes in both amide carbonyl and amino groups of doxycycline. In fact, the peaks related to the amide C=O were shifted to 1674 cm^−1^, while the –NH_2_ shifted from 1525 to 1533 cm^−1^, suggesting an interaction between the polymeric blend and the drug via its amide group.

### 3.2. In Vitro Drug Release

In vitro permeation experiments were conducted using Franz-type vertical diffusion cells and PBS as the acceptor fluid to assess the release behavior of the encapsulated drug. Figure 6 exhibits representative release curves of doxycycline from APC particles produced with different polysaccharides ratios.

All formulations showed a burst effect in the first hours (4 h) after administration followed by a prolonged release over the next hours or days related to the formulation. Such a burst in release supported by the slow, sustained delivery of the antimicrobial agent can be very useful to avoid the spreading of bacterial contamination and the treatment of different wound infections, promoting the wound healing process [63]. As shown in Figure 6, pure doxycycline was completely released in less than 15 min, whereas APC-111-2D showed a 60% drug release in 6 h and a complete release of doxycycline within 72 h, while powders containing a higher amount of chitosan, APC-113-2D and APC-117-2D, released at the same time about 40% and 35% in 6 h, and released 50% and 65% at 72 h, respectively. In spite of having a smaller size, the total release of APC-113s and APC-117s formulations was achieved in 5 and 7 days, depending on the amount of chitosan in the formulations. Prolonged release of the drug in formulations with a higher amount of chitosan can be related to both interactions between doxycycline and chitosan, as demonstrated by FT-IR studies, and higher consistency of the hydrogels. In fact, rheology studies focused on tan δ, the factor that represents the ratio between viscous and elastic behavior, highlighted an increase in the viscoelastic behavior of the hydrogels according to the amount of chitosan. Usually, in the practical applications, 0.01 < tan δ < 1 describes a gel-like behavior, where 0.5 represents the transition point from a prevalent solid-like to a prevalent liquid behavior. In the AP formulation, tan δ is >0.5, on the contrary, in APC formulations, this value decreases proportionally to the increase of chitosan concentration, demonstrating that in this kind of hydrogels, the corpuscular part predominates over the liquid part (see Appendix A). As shown in Appendix A, APC-117 showed a *G*′ (storage modulus) about 10% higher than APC-113 and 18% higher than APC-111. Formulations loaded with a lower amount of doxycycline, namely 1% (*w*/*w*), did not exhibit a significant difference compared with the homologous formulations loaded with a higher dose (data not shown), confirming that gel properties are the main driving factors of the release.

### 3.3. Powders Antimicrobial Activity

With the aim to test the effect of the technological process on doxycycline antibiotic activity, a preliminary antimicrobial assay was carried out, using a modified disc diffusion test conducted on a Muller–Hinton agar plate spread with *S. aureus*, which normally colonizes the wounds [64]. Figure 7 shows an inhibition zone with a diameter of 30.64 mm for pure doxycycline, while APC drug-loaded powders show an inhibition diameter of 29.87, 32.01, and 32.86 mm, for APC-111-2D, APC-113-2D, and APC-117-2D, respectively. Moreover, a time-kill assay measuring the reduction of *S. aureus* recovered at 1, 2, 3, and 7 days was conducted. A strong reduction of *S. aureus* compared to the control was found for both drug and drug-loaded powders on day 1. Pure doxycycline lost its activity after 2 days, whereas all drug-loaded APCs formulations retained their activity longer, with APC-117-2D able to retain most of the antimicrobial activity over the 7-day experiment due to the prolonged release of the drug. The increase in antimicrobial activity related to the chitosan amount into the powders can also be explained considering the bacteriostatic effect of chitosan [65], as demonstrated by blank formulation APC-117.

### 3.4. In Vitro Test

#### 3.4.1. MTT and ELISA Assay

Biocompatibility of the polysaccharides blend powders was assessed using human keratinocyte HaCaT monitoring cytotoxic activity (MTT assay). HaCaT cells were exposed to single polymers alginate, pectin, and chitosan, to two polymers blend in different combinations, and alginate/pectin/chitosan blends at different ratios (1:1:1, 1:1:3, and 1:1:7). As shown in Figure 8A, no formulation made any of the two or three polymers blend produce any significant cytotoxic activity in the range of 0.01–10 µg/mL. 

The toxicity test results were used to set the range of concentrations (0.1–0.5–1 µg/mL) to evaluate the pro-inflammatory activity of the powders by the release of the cytokines IL-6 and TNFα and of the chemokine IL-8 (CXCL-8) mediators in the inflammatory process [66,67].

As expected, particles made by the single polysaccharides did not show significant pro-inflammatory activity in terms of the release of cytokines in human keratinocytes, such as IL-6 and TNF-α, and chemokine, such as IL-8. As shown in Figure 8B, APCs did not alter the levels of IL-6 and TNF-α compared to their basal levels. Interestingly, APC formulations were able to induce, even within the limits of significance (*p* = 0.04), a higher release of IL-8 from the human keratinocytes when tested at the concentration of 1 µg/mL. This result might be due to the interaction between chitosan and alginate that led to the reduction of positive charge and the increase of hydrophobicity of the chitosan. Consequently, the chitosan interacting with cells might favor the release of IL-8 [68]. As reported in the literature, IL-8 is able to increase the keratinocytes migration rate suggesting that the APC powders could stimulate the wound healing process in difficult-healing wounds [69].

#### 3.4.2. SDS-PAGE Gelatin Zymography

Gelatinases (Metalloproteinases 2 and 9) are involved in regulating the cell migration essential for wound reepithelialization, but uncontrolled proteolysis can lead to a delay of healing [70]. For this reason, a decrease in MMPs is correlated with wound healing [71]. Clinical and experimental studies reported that doxycycline, at low doses, is able to inhibit MMP-2 and 9, reducing both the expression and activity of these proteases [72,73,74,75,76].

To study the activity of the powders loaded with doxycycline on the MMPs, human epidermal keratinocyte model cells were placed in trans-wells and treated with a hydrogel containing different dox amounts, avoiding any direct contact with cells. After 24, 48, and 72 h of incubation, the culture media was subjected to gel zymography. The gels showed that only MMP-9 (as deduced from the molecular weight) was detected in HEKa conditional media and that doxycycline released from hydrogel inhibited its activity even at 0.5 μg/mL, and its effect was stable up to 72 h. Interestingly, drug-loaded APC powders were more active than pure doxycycline at any tested concentration, confirming that drug encapsulation did not alter the bioactivity of doxycycline but was able to enhance it probably due to the prolonged release of the drug compared to administration of the drug in the form of a solution, as shown in Figure 9.

## 4. Conclusions

Spray drying technology has been successfully used to produce in situ gelling powders made of either alginate/pectin or alginate/pectin/chitosan blends loaded with doxycycline, an antimicrobial model drug with a wide spectrum and the ability to inhibit host matrix metalloproteinases, hyper expressed in chronic wounds. Size, morphology, and fluid uptake ability of the particles were influenced by the presence of chitosan, as well as the drug encapsulation efficiency, between 66% and 78%. 

Release in SWF of the encapsulated doxycycline from the in situ-formed hydrogel exhibited a burst effect within 4 h, followed by a prolonged release over 24 h, depending on the alginate/pectin/chitosan ratio, with the slower release related to a higher amount of chitosan due to an extensive interaction between the polymeric blend and the drug via the amide group, as demonstrated by FTIR studies. Such an interaction did not affect the antimicrobial activity of doxycycline, which was even enhanced in drug-loaded particles with a high amount of chitosan. Moreover, in situ gelling powders did not show pro-inflammatory activity in terms of the levels of IL-6 and TNF-α, whereas they were able to induce the release of IL-8 from the human keratinocytes that could stimulate the wound healing process in difficult healing. Interestingly, doxycycline-loaded particles were able to increase drug activity against MMPs, with good activity against MMP-9 even at 0.5 μg/mL after 72 h. Such results suggest that in situ gelling powders rich in chitosan could be a promising wound dressing.

## 5. Patents

Del Gaudio P, Aquino R.P, Russo P, De Falco G, Nicolais G. “In situ gelifying powder” PCT/IB2018/058742 (Worldwide database Pub num = WO/2019/092608, Publication Date: 16 May 2019; International Filing Date 7 November 2018).

## Figures and Tables

**Figure 1 pharmaceutics-13-01680-f001:**
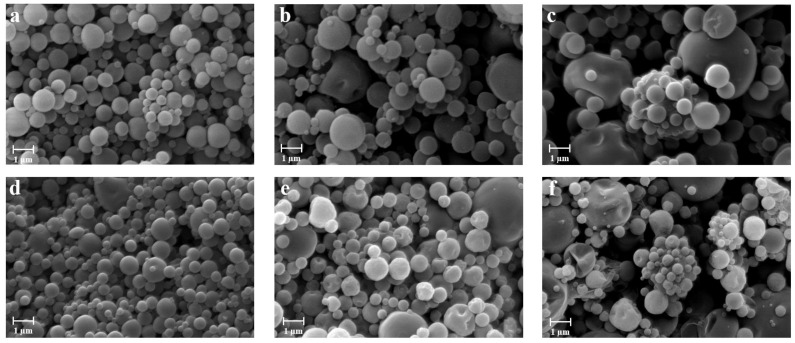
SEM microphotographs of alginate-pectin particles produced by mini spray drying with different polymers ratios: blank particles AP-11 (**a**), AP-31 (**b**), AP-13 (**c**), and doxycycline-loaded particles AP-11-2D (**d**), AP-31-2D (**e**), AP-13-2D (**f**).

**Figure 2 pharmaceutics-13-01680-f002:**
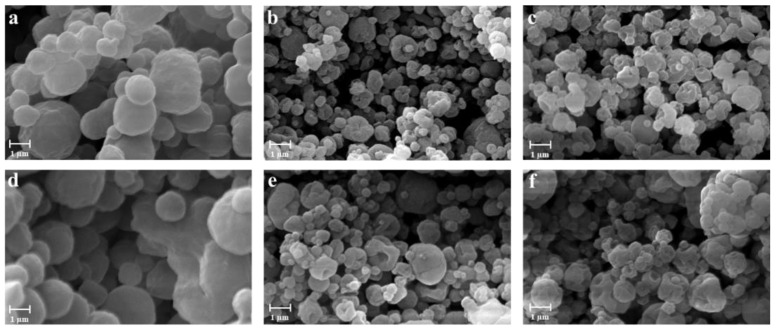
SEM microphotographs of alginate-pectin particles produced by mini spray drying with different polymers ratio: blank particles APC-111 (**a**), APC-113 (**b**), APC-117 (**c**), and 2% (*w*/*w*) doxycycline-loaded particles APC-111-2D (**d**), APC-113-2D (**e**), APC-117-2D (**f**).

**Figure 3 pharmaceutics-13-01680-f003:**
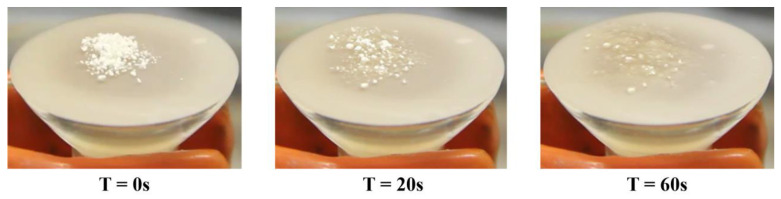
Simulated wound fluid uptake of in situ gelling powder APC-117.

**Figure 4 pharmaceutics-13-01680-f004:**
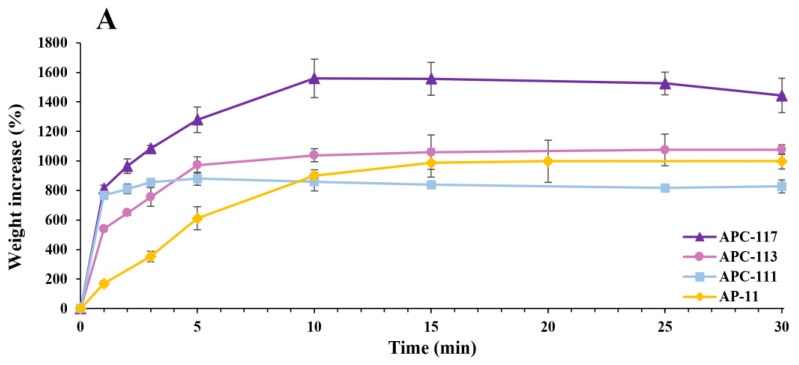
Simulated wound fluid uptake of in situ gelling powders: Panel (**A**): blank alginate/pectin/chitosan particles in comparison with alginate/pectin particles. Panel (**B**): alginate/pectin/chitosan powders loaded with different amounts of doxycycline.

**Figure 5 pharmaceutics-13-01680-f005:**
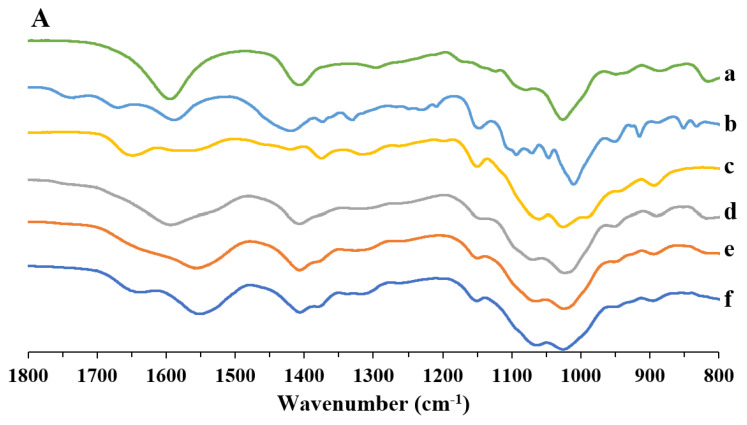
FTIR spectra alginate/pectin/chitosan particles in comparison with doxycycline-loaded polymeric particles and drug raw material. Panel (**A**): high M content alginate particles (a), amidated pectin particles (b), low MW chitosan particles (c), and APC-111 (d), APC-113 (e), APC-117 (f). Panel (**B**): doxycycline raw material (g), APC-111 (h), and APC-111-2D (i).

**Figure 6 pharmaceutics-13-01680-f006:**
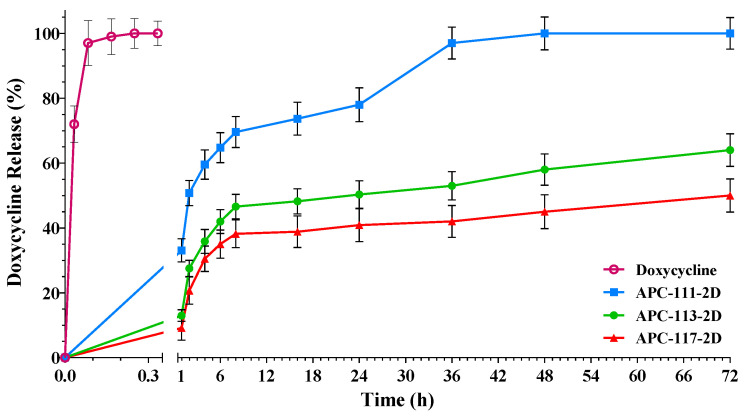
Release profiles of doxycycline-loaded alginate/pectin/chitosan powders with different polymer ratios.

**Figure 7 pharmaceutics-13-01680-f007:**
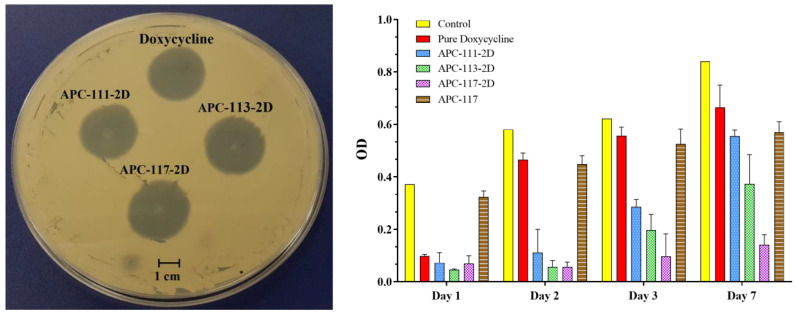
Modified diffusion assay conducted against *Staphylococcus aureus* at 24 h by APCs-2D at different ratios in concentrations equivalent to 1.55 µg of pure doxycycline, used as control, on the left; on the right, time-kill assay against *Staphylococcus aureus* obtained by detaining the wells with acetic acid and measuring the absorbance of the CV at 595 nm.

**Figure 8 pharmaceutics-13-01680-f008:**
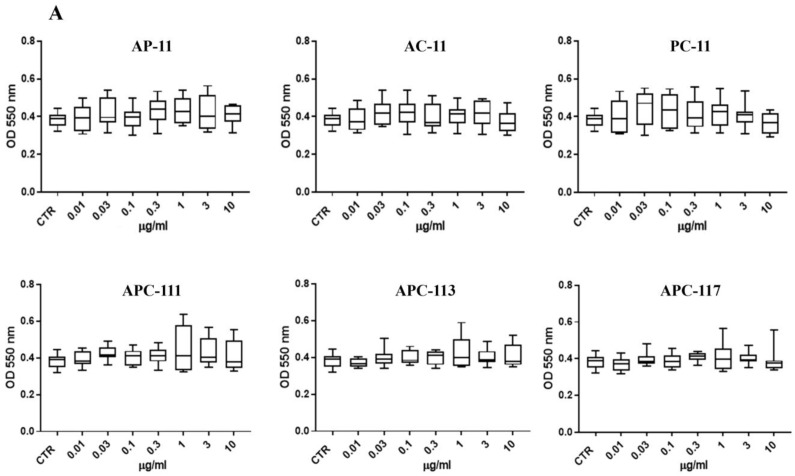
Bioavailability of in situ gelling powders made by different polysaccharide combinations. Panel (**A**): Cell viability assessed by MTT test on HaCaT cells after 24 h of treatment with two polymer blends alginate/pectin, alginate/chitosan, pectin/chitosan, and three a polymer blend, APC, with different polysaccharide blends. Data are represented as median ± interquartile range (*n* = 7). The statistical analysis was determined by one-way ANOVA with the Bonferroni test as a post-test. Panel (**B**): Pro-inflammatory effect of APCs particles at the various concentrations in terms of release of IL-6, TNFα, and Il-8. The statistical analysis, obtained through the Two-Way ANOVA followed by Tukey’s post-test, was carried out with respect to the basal levels.

**Figure 9 pharmaceutics-13-01680-f009:**
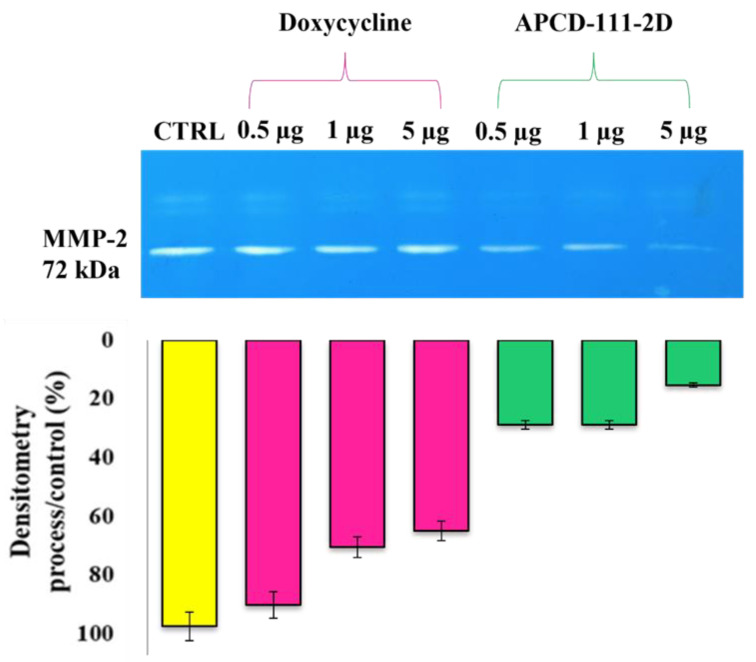
Representative zymograms of pure doxycycline and APC-111-2D. Incubation of the hydrogels containing 0.5, 1, and 5 µg/mL of doxycycline at 72 h.

**Table 1 pharmaceutics-13-01680-t001:** Composition, process yield, particle size, drug content, and encapsulation efficiency (E.E.) of powders obtained with different polysaccharides ratio by mini spray drying.

Sample Code	Alginate/Pectin/Chitosan Ratio	Doxycycline (*w*/*w*)	Yield (%)	Drug Content (%)	E.E. (%)	Mean Diameter (µm)
AP-11	1:1	-	63	-	-	2.07 ± 0.08
AP-31	3:1		65			2.16 ± 0.11
AP-13	1:3		61			2.85 ± 0.49
AP-11-1D	1:1	1.0%	62	0.70	69	2.11 ± 0.12
AP-11-2D	1:1	2.0%	62	1.41	70	2.13 ± 0.15
AP-31-2D	3:1		64	1.33	66	2.21 ± 0.13
AP-13-2D	1:3		66	1.38	69	2.74 ± 0.43
APC-111	1:1:1	-	60	-	-	3.24 ± 0.13
APC-113	1:1:3		72			2.75 ± 0.10
APC-117	1:1:7		73			2.58 ± 0.04
APC-111-1D	1:1:1	1.0%	62	0.69	69	3.25 ± 0.08
APC-113-1D	1:1:3		68	0.72	72	2.78 ± 0.09
APC-117-1D	1:1:7		70	0.78	78	2.24 ± 0.13
APC-111-2D	1:1:1	2.0%	62	1.33	67	3.59 ± 0.09
APC-113-2D	1:1:3		71	1.41	71	2.69 ± 0.12
APC-117-2D	1:1:7		74	1.56	78	2.42 ± 0.11

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
