# Peer review of "A Novel Three-Polysaccharide Blend In Situ Gelling Powder for Wound Healing Applications"

_pharmaceutics, 2021, doi:10.3390/pharmaceutics13101680_

Round 1

Reviewer 1 Report

Dear Authors,

This paper is well written, and the experimental part is sustained by complex physico-chemical, biopharmaceutical and biological analysis.

I have some minor remarks related to:

  1. I recommend Authors, for further studies related to pharmaceutical powders, to characterize them from goniometric point of view, determing the contact angle as an indicator of their hydrophilicity.
  2. In my opinion, in Figure 5, is more correct to put doxycycline cumulative released (%) instead doxycycline permeation (%).

This paper has important perspectives in the development of local formulations as in situ gelling powders for wound healing applications.

Further clinical studies are required for using the new designed topical formulations as wound dressings.

Some minor English revision and spelling corrections are required.

Author Response

1. I recommend Authors, for further studies related to pharmaceutical powders, to characterize them from goniometric point of view, determining the contact angle as an indicator of their hydrophilicity.

Thank you very much for the suggestion. Surely in the next phase of powder development contact angle will be taken into account to better characterize the in situ gelling powder

2. In my opinion, in Figure 5, is more correct to put doxycycline cumulative released (%) instead doxycycline permeation (%).

Thank you again for the suggestion. The axis title has been properly modified as suggested

3. Some minor English revision and spelling corrections are required.

English has been revised in order to avoid some typos 

Reviewer 2 Report

The authors prepared in situ gelling microparticle powders composed of alginate, pectin, chitosan and doxycycline for would healing. The powders were prepared by spray drying method, and single micron size of particles were obtained. They used combination of alginate, pectin, and chitosan to prepare microparticles, and investigated the influence of chitosan and the inclusion of chitosan affected the drug release and weight increase. They investigated the cytotoxicity and cytokine secretion in vitro. Overall, I agree the manuscript for the publication of this journal.

-Figure 3B: axis is typo

-How long does the formulation complete the drug release (Fig. 5)? Are the release form microparticle incomplete?

-HaCaT cells may be not appropriate for the analysis of biocompatibility…

-Can you show any information about particle distribution in supplementary Figure or Table for the kindness to readers?

-Did the particle size affect the drug release and weight increase?

-5. Patents

Author Response

All the authors want to thank the reviewer for the suggestions that would increase the quality of the paper. 

About your questions:

1. Figure 3B: axis is typo

The typo in figure 3B as others in the manuscript have been corrected

2. How long does the formulation complete the drug release (Fig. 5)? Are the release form microparticle incomplete?

Information about doxycycline total release has been included into the manuscript. Moreover, figure 5 has been edited in order to enhance the differences in release between the various drug loaded powder. 

3. HaCaT cells may be not appropriate for the analysis of biocompatibility…

Although there are specific cell lines used in biocompatibility studies, HaCaT are extensively used in many studies and accepted at large as reliable model for both biocompatibility and inflammatory response, as reported in literature (HaCaT Cells as a Reliable In Vitro Differentiation Model to Dissect the Inflammatory/Repair Response of Human Keratinocytes. Mediators Inflamm. 2017. doi: 10.1155/2017/7435621; Evaluation of Biocompatibility and Cytotoxicity Using Keratinocyte and Fibroblast Cultures. Skin Pharmacology and Physiology. 2009, doi: 10.1159/000178866)

4. Can you show any information about particle distribution in supplementary Figure or Table for the kindness to readers?

A table with data about d10, d50 and d90 has been added as supplementary material to better characterize the particle size distribution

5. Did the particle size affect the drug release and weight increase?

Manuscript has been re-edited in several parts to highlight the effect of particles size on both particles fluid uptake properties and drug release behaviour

6. Patents

The code and other information regarding the patent have been highlighted